# Integrating Science Media Literacy, Motivational Interviewing, and Neuromarketing Science to Increase Vaccine Education Confidence among U.S. Extension Professionals

**DOI:** 10.3390/vaccines12080869

**Published:** 2024-08-01

**Authors:** Erica Weintraub Austin, Nicole O’Donnell, Pamela Rose, Zena Edwards, Anya Sheftel, Shawn Domgaard, Di Mu, Paul Bolls, Bruce W. Austin, Andrew D. Sutherland

**Affiliations:** 1Edward R. Murrow College of Communication, Washington State University, Pullman, WA 99164, USA; nicole.odonnell@wsu.edu (N.O.); pamela.rose@wsu.edu (P.R.); di.mu@wsu.edu (D.M.); pbolls@wsu.edu (P.B.); andrew.sutherland@wsu.edu (A.D.S.); 2Extension Youth and Families, Washington State University, Vancouver, WA 98686, USA; zena_edwards@wsu.edu; 3Teaching and Learning, Washington State University, Pullman, WA 99164, USA; anya.sheftel@wsu.edu; 4Department of Communication, Hawaii Pacific University, Honolulu, HI 96813, USA; sdomgaard@hpu.edu; 5Kinesiology and Educational Psychology, Washington State University, Pullman, WA 99164, USA; bwaustin@wsu.edu

**Keywords:** associated factors, COVID-19, trust, willingness, vaccination

## Abstract

This article presents an Integrative Model of Sustainable Health Decision-Making and a toolkit to equip U.S. Extension professionals with knowledge and skills to engage in adult immunization education. The objective was to reduce mistrust and increase willingness and confidence toward delivering vaccination education. The model was developed through an explanatory parallel mixed methods design. Data collection included a needs assessment survey, interviews, workshops, and Neuromarketing message testing. The resulting toolkit was pilot tested before final delivery. Four key needs were identified: tailoring trainings based on Extension roles, prioritizing preserving community trust and professional credibility, establishing connections with medical experts, and strengthening Science Media Literacy skills to counter misinformation and communicate emerging science. Correlations among constructs supported an integrated model focused on a professional development core of Science Media Literacy, Motivational Interviewing, and Neuromarketing Science that strengthens communication relationships between priority populations and trusted partners. The model and work described in this article can serve as a general framework for engaging key influencers in communities in communication education intended to promote sustainable well-being, such as increasing vaccine uptake.

## 1. Introduction

The COVID-19 pandemic has had a significant negative global impact on health, with disproportionate effects in the U.S. on rural and immigrant communities, as well as communities of color [1,2,3]. Furthermore, these communities exhibited higher levels of vaccine hesitancy [4], fueled in part by misinformation about vaccine safety [5], lack of trust in vaccine safety, and lower levels of concern about the seriousness of COVID-19 [6]. In June 2021, the United States government allocated additional funds to the Extension Collaborative on Immunization Teaching and Engagement (EXCITE), which was already addressing vaccination-related health disparities among priority populations (e.g., rural communities and communities of color). Through the initiative based on this funding, Cooperative Extension professionals engaged in evidence-based adult immunization education in their local communities. This project addresses how a community-based needs assessment investigating determinants of trust, hesitancy, and willingness facilitated the development of a successful professional development program to facilitate improved project success to promote adult vaccine education, including for COVID-19. 

U.S. Extension professionals hold a unique and trusted position in their communities, enabling them to bridge gaps in clinical resources, education, and preventative services for disease prevention [7,8]. The Cooperative Extension system, established through the Smith–Lever Act of 1914, utilizes land grant universities to translate academic research into practical knowledge that benefits community socioeconomic functioning [9]. Since 2014, Cooperative Extension has also prioritized health equity focusing on equitable access to health and well-being services by promoting health behaviors through communication and education and working with state and national coalitions to address social determinants of health (e.g., employment, housing, food access, education, and social connectedness) [10,11]. This has included childhood immunization education with parent and early childhood education. Despite its recent focus on health equity, some Cooperative Extension offices were hesitant to partner with the EXCITE initiative. This hesitancy stemmed from various factors, including low confidence for success, limited time and capacity, staff shortages, lack of training to provide vaccine education, resistance in some communities, and low willingness to engage in adult COVID-19 vaccine education [12]. 

To address these challenges and decrease Extension professionals’ hesitancy to provide vaccine education, EXCITE funded a team of Media Literacy and communication experts at Washington State University (WSU). This team was tasked with (1) conducting a comprehensive needs assessment of Extension professionals’ hesitancy, willingness, and confidence to engage in adult vaccine education and (2) developing an educational intervention aimed at increasing Extension professionals’ skills, willingness, and confidence in this area. This manuscript describes the results of a community-based needs assessment investigating determinants of trust, hesitancy, and willingness among U.S. Extension professionals to promote adult vaccine education, including for COVID-19.

### 1.1. Literature Review

#### 1.1.1. Rationale for Approach to Needs Assessment for Addressing Vaccine Hesitancy 

The importance of emotional processes in vaccine science education was the basis for the development of the needs assessment. Previous research has highlighted the importance of considering psychological reactance [13], an emotionally defensive response to a perceived threat to one’s freedom, which health communicators need to guard against [14]. However, skills like media literacy are strong predictors for COVID-19 protective health behaviors. Media literacy refers to the understanding that every media message is constructed for a specific goal and uses medium-/context-specific techniques to influence beliefs, attitudes, values, and behaviors of media consumers [15]. Austin et al. [16] found that media literacy and trust in expert sources such as the Centers for Disease Control, the World Health Organization, the National Institute of Infectious Diseases, and local health departments together predicted more variance in COVID-19 vaccine intentions than knowledge about COVID-19 impacts. Furthermore, Austin and colleagues found that consumer expectancies of immunization behaviors—the “if-then” beliefs about the possible benefits of performing a promoted behavior—directly impacted consumer willingness to engage in this behavior. Similar results are present in the international arena [16]. For example, the results from a survey conducted in 126 countries indicated that, in countries with high levels of trust in science, individuals were more vaccine confident [17]. Similarly, the results of a large-scale survey in Asian and Western countries indicated that trust in government decreases the impact of misinformation on vaccine uptake [18]. This points to the complicated affective environment of vaccine science education. Traditional cognitive processes such as knowledge take a backseat to affective processes driven by emotion and affective cognitions such as trust, willingness, expectancies, and efficacy. Given this emotionally charged context, it is understandable that some Cooperative Extension professionals would express unwillingness to engage in vaccine science education in their communities. However, the Cooperative Extension professionals are well suited to deliver an impactful vaccine education, since they can leverage community partnerships and have built trusting relationships with community members through other Extension initiatives (e.g., agricultural education, nutrition, and youth development), characteristics that can support vaccine education initiatives [19]. Below, we describe the elements of the needs assessment to further understand the Cooperative Extension professionals’ perspectives and develop a set of educational materials that could increase their confidence in delivering vaccine education to priority populations.

#### 1.1.2. Elements of the Needs Assessment: Science Media Literacy

In an expansive and confusing media environment, individuals need to decipher between accurate and inaccurate information. Unfortunately, people often overestimate their media literacy abilities and can be misled by emotive but unrealistic or inaccurate content. This can lead to the sharing of false information and mistakes in evaluating health information [20,21,22,23]. Fortunately, research has found that individuals who engage with Science Media Literacy have more correct knowledge about COVID-19 [24], and studies have shown consistently that media literacy-oriented interventions can boost resistance to misinformation and receptivity to health campaigns [25,26,27,28]. Misinformation beliefs can also arise from a perceived dearth of trusted, locally based, and culturally connected sources of health information [29], which can lead to the acceptance of messages from appealing but misleading social media-based sources [30]. Because some sources attempt to use trust to manipulate media users through partisanship, emotions, and other psychological factors, media literacy tools can address misinformation while also promoting trust with others. A communication strategy that applies and integrates Science Media Literacy with other evidence-based strategies to promote behavior changes can therefore help Extension professionals to build trusting relationships with community members while promoting well-being and specific health decisions that address misinformation.

#### 1.1.3. Elements of the Needs Assessment: Motivational Interviewing

Motivational Interviewing (MI) is an evidence-based approach to support health behavior change [31,32] and, more recently, has been shown to be a promising intervention to support immunization-related behaviors [33,34,35]. The core tenet of MI is that, by creating a positive alliance and building trust with community members, health professionals can effectively support community members in exploring their own reasons for change and reinforce these reasons by using MI strategies, thus increasing the likelihood of community members initiating and maintaining behavior change [31,36,37]. Additionally, MI supports the evocation of positive emotions such as hope and empowerment [38], which can reduce reactance to novel information and behaviors [39]. 

MI has been used as an effective public health intervention to support physical and behavioral health outcomes among priority populations [40,41,42]. Furthermore, over the past decade, the utility of MI within Cooperative Extension has gained interest [43,44]. For example, Parisi and colleagues [45] demonstrated the positive impact of a MI-informed Hypertension Management Program used by Extension professionals on community members’ blood pressure. Within the context of our EXCITE needs assessment, the project team hypothesized that equipping Extension professionals with MI skills would equip them with tools to (1) reduce negativity response spirals and optimize resonance with COVID-19 vaccine-related education and outreach, thus (2) increasing their willingness and confidence in engaging in COVID-19 vaccine education with priority populations. The project team also hypothesized that the use of MI by Extension professionals to strengthen trust and positive relationships with priority populations would reduce the boomerang effect among priority populations—or the rejection of information contrary to one’s beliefs [46]—and support these populations in using Science Media Literacy skills to seek out credible sources of information [47].

#### 1.1.4. Elements of the Needs Assessment: Neuromarketing Science

The field of Neuromarketing Science refers to the systematic study of how the human mind—embodied in the nervous system—processes and responds to media content and technology [48]. Neuromarketing Science consists of two related yet independent specialized areas. The first area of Neuromarketing consists of a body of knowledge about how the human mind—embodied in the brain—works and processes lived experiences such as being exposed to vaccine education content. This body of knowledge, built on over 40 years of scientific research on mental processes underlying media effects, provides actionable insights into producing all forms of effective media content, including vaccine science messaging in support of vaccine therapy. The field of Neuromarketing Science also contains a body of formal measures and methods for conducting primary research. Neuromarketing Science therefore served as a rigorous element of the needs assessment. Both areas of Neuromarketing Science were applied to this specific project.

Neuromarketing Science has an indirect, although crucial, role to play in vaccine therapy. The benefits of vaccine therapy are ideally realized through voluntary, informed health decisions by individuals on a mass scale. This requires extremely effective communication and messaging about vaccines and vaccine science in support of advancing vaccine therapy. This is especially challenging in the current highly emotional and politicized communication environment surrounding the topic of vaccine therapy, including, but also reaching beyond, COVID-19 vaccination. Neuromarketing Science therefore provides a scientifically rigorous and practically powerful approach to informing the production and delivery of critical communication content about the importance of vaccines and vaccine therapy. 

The particular role that Neuromarketing Science plays in vaccine therapy is to optimize communication content strategies that can empower individuals to make science-based decisions concerning the acceptance or rejection of vaccine therapy. Neuromarketing is an approach to testing content production that can optimize messaging about vaccines by preventing a negative cascade of emotions that hinder effective health decision-making. This makes possible a communication environment that facilitates the cognitive processing of vaccine science information and supports informed health decision-making that distinguishes accurate scientific information from misinformation. 

This process is how Neuromarketing Science can play an indirect role in vaccine therapy that directly supports health decision-making, which is required in vaccine adherence. Informed health decision-making should be the goal of all messaging about vaccination and vaccine adherence. Facilitating informed decision-making in an emotionally fraught environment is especially crucial given the complexities of vaccine science that encompass highly complex topics such as the difference between a life-challenging vaccine pathogen and a routine vaccine used as preventative medicine. This is what made Neuromarketing Science a particularly useful tool for the needs assessment to produce messaging most likely to be effective in the scientifically complex, highly emotional health communication environment faced by the EXCITE team. 

The value of Neuromarketing Science, as a field and conceptual foundation for this project, has been demonstrated repeatedly in its history of providing actionable insights to increase the effectiveness of content for health communication, advertising, news, entertainment, and politics spanning communication and media channels [49]. Lee et al. [50] recently outlined how this approach can inform strategic communication to engage organizational stakeholders such as Extension professionals and community members through what is termed, in this project, brain-friendly content. 

The application of Neuromarketing Science to the needs assessment for this project was two-pronged. The needs assessment involved conducting qualitative/interpretive primary research using Neuromarketing measures and methods. This consisted of a designed experiment testing the impact of vaccine visual cues and message framing on responses to COVID-19 vaccine messaging. This was a small sample study grounded in the qualitative directional interpretation of data rather than statistical significance testing. Valid techniques for data analysis in Neuromarketing Science encompass projects using statistical significance testing and projects only using qualitative directional data analysis and interpretation [51]. The application of Neuromarketing Science in this project also focused on developing content for the toolkit grounded in the accumulated body of existing Neuromarketing knowledge focused on how the human mind processes and responds to media content and technology. This is a body of knowledge that has been growing since the 1980s [52] and can be used to produce practical insights for communication professionals that guide the production of what is termed brain-friendly content—content optimized for how the human mind processes media. The practical insights that emerge from Neuromarketing Science are rooted in formal principles.

Neuromarketing Science principles are largely guided by theoretical assumptions from Lang [53,54] about how the human mind processes and responds to communication and media content. Three of the most important facts guiding the application of Neuromarketing Science for this project include:The effects of media entirely emerge from cognitive and emotional processes embodied in the human brain and nervous system.Humans are, first and foremost, motivational and emotional information processors.Humans are limited capacity information processors.

This means that, when individuals experience and interact with media content and technology, they infer meaning through a process that involves cognition and emotion, which always involves bottom-up (emotion-led) processing. This provides the potential for maximizing positive emotions when interacting with media content, thus reducing threat perceptions and increasing openness to novel information. It also provides the potential for negative emotions to decrease receptivity to new information.

The WSU-EXCITE needs assessment helped frame the context for, and was used to directly inform, our specific application of Neuromarketing Science in this project. Our specific application of Neuromarketing Science encompassed equipping Extension professionals with communication expertise grounded in this approach, as well as the application of Neuromarketing methods through applied COVID-19 vaccine content testing. Independent of the survey and focus groups, it was theorized that developing content for the toolkit grounded body of knowledge tied to Neuromarketing Science could provide Extension professionals with foundational knowledge that should empower them to communicate more effectively about emotionally charged topics like COVID-19 vaccination. This theorizing was grounded in the history of Neuromarketing for optimizing content. Equipping Extension professionals with communication expertise focused on developing the toolkit content intended to train Extension professionals in what is termed brain-friendly content. Developing brain-friendly content requires grounding content development, design, and delivery in facts about the human brain/mind [55]. 

#### 1.1.5. Summary of the Needs Assessment Elements

This project therefore integrated the application of theories and methods from Science Media Literacy, Motivational Interviewing, and Neuromarketing Science. A challenge faced by EXCITE that was supported by the needs assessment is getting Extension professionals to embrace human vaccine science education as part of the Extension mission. An insight gained through the results of the needs assessment survey and focus group is that framing the mission of COVID-19 vaccine education within the context of Extension values [56] could resonate with Extension professionals and potentially reduce resistance to this new focus.

## 2. Materials and Methods

The needs assessment comprised the explanatory parallel mixed methods design to explore Extension professionals’ attitudes and beliefs about engaging in COVID-19 education with priority populations [57]. The purpose of this design was to deepen project teams’ understanding of Extension professionals’ hesitancy to provide COVID-19 vaccine education and outreach by first collecting quantitative survey data, followed by focus groups with Extension professionals and in-depth interviews with Cooperative Extension administrators. 

This was followed by empirical applied Neuromarketing content testing of ways of framing the topic of COVID-19 vaccine education within specific Extension values. This was done using a portable Neuromarketing lab at two Extension professional conferences, one of which attracted a particularly diverse set of participants. A strength of Neuromarketing methods is the ability to test how individuals unconsciously and consciously mentally process and respond to content. The research methods of Neuromarketing Science holistically combine physiological indicators of cognitive and emotional processes with self-report and behavioral measures to gain valid and practically valuable insights into how individuals mentally process and respond to content [51,52].

As shown in Figure 1, the data collection tools included a quantitative survey, six focus groups with Extension professionals, 10 in-depth interviews, and two sets of Neuromarketing-based message testing sessions convened at two professional conferences based on the vaccine promotional message concepts that emerged from the needs assessment. Pilot testing of concepts for the professional development tool using zoom workshops led to further zoom-based pilot testing of the final toolkit before launching it. The project was approved as exempt by the Washington State University Institutional Review Board (#19380).

### Survey Methods

The WSU project team developed a quantitative survey that collected data about Extension professionals’ (1) demographics, (2) professional identity, (3) acceptance of Extension doing vaccine education, (4) willingness and confidence to do vaccine education, (5) comfort in managing the misinformation environment, (6) Science Media Literacy, (7) perceptions about COVID-19 risks, (8) trust in information and sources, (9) social identity, and (10) willingness to speak out. Questions were adapted from previous Extension assessment tools, CDC rapid community assessment tools, previously published studies by the team, and Science Media Literacy measures adapted from Austin et al. [24] for this study. The survey also included open-ended questions that asked about what would positively impact respondents’ willingness and confidence to engage in COVID-19 vaccine education, as well as opportunities to provide any other comments. At the end of the survey, respondents were asked to provide their email address if they were willing to participate in a focus group. A WSU Extension educator led the survey design based on recent Extension surveys, established measurement constructs, communication theory, and the Total Design Method [58]. 

In Spring 2022, the WSU project team invited Extension professionals to complete a quantitative survey. Participants were recruited via email using the Extension management service’s email and newsletter distribution lists. Initial recruitment efforts resulted in 1341 professionals starting the survey, with a 75.26% completion rate and a final sample of *n* = 1009 participants. Most of the respondents were women (71.9%). Individuals could mark more than one race/ethnicity category, and the survey results indicated that respondents identified as White (*n* = 603), Black or African American (*n* = 70), Hispanic or Latino (*n* = 28), Native American or Alaska Native (*n* = 14), Asian (*n* = 8), Native Hawaiian or other Pacific Islander (*n* = 4), and other/prefer to self-describe (*n* = 10). Respondents were asked to identify in which of the five Cooperative Extension regions they worked; the results included regional representations from the Southern (38.5%), Western (19.5%), North Central (19.0%), Northeast (16.7%), the 1890 institutions region, which refers to a group of 19 Historically Black Colleges and Universities (HBCUs) (3.7%), and missing (2.7%) regions. Respondents indicated their Cooperative Extension program areas as Family and Consumer Sciences (25.2%), Agriculture (23.2%), 4-H Youth Development (18.3%), Natural Resources (5.3%), Community Development (5.2%), other (22.3%), and missing responses (0.6%). Note, some Extension professionals worked in more than one program area. Our team aimed to reach a sample of Extension professionals that would be representative of the organization, including individuals from various geographic areas and Extension roles. Additionally, our sample had a higher proportion of females, reflecting the organization’s overall employee sex distribution [59].

Science Media Literacy, a newly created construct for this study, was measured using an updated version of Austin et al.’s [24] construct to better emphasize science information versus the scientific method, as they had recommended [60], consistent with McClune and Jarman’s [61] five categories of knowledge and skills for critically consuming science news, the National Association for Media Literacy Education [15] core principles, and Bergstrom et al.’s [62] recommended skills for understanding science as a competent outsider.

## 3. Results

### 3.1. Survey Results

The needs assessment survey identified key insights into the most pressing resources that professionals need to make an informed choice in their own best interests and to participate as active agents in vaccine education in their communities, and they revolved around four needs.

#### 3.1.1. Need 1. Tailor Training Based on Extension Roles 

Vaccine education willingness and confidence to implement programming varied based on Extension roles and areas of expertise. Extension leaders and those who worked in Family and Consumer Sciences and Community Development expressed the highest levels of willingness and confidence. These Extension professionals may have more experience with vaccine education, which could shape their responses to the survey, as well as how their brains process the specific content that was tested later through Neuromarketing. In terms of regions or designations, individuals in 1890 institutions indicated the highest levels of willingness and confidence. These results indicated that educational materials should be created to support the unique needs and perspectives among the diverse specialties that give Extension such strength across its many regions (Figure 2).

#### 3.1.2. Need 2. Prioritize Preserving Community Trust and Professional Credibility 

Extension professionals across our sample expressed concern over diminishing trust within their communities. This included 26% who reported that stakeholders had expressly disapproved of Extension facilitating COVID-19 vaccination-related education, outreach, and/or programming. As well, among those participating in vaccination education, 17.1% indicated that they had been treated poorly by others during the COVID-19 pandemic because they are a vaccination educator. Extension professionals will likely benefit from having the autonomy to adapt training to address community needs based on their local expertise. The importance of trust came up unsolicited in 40 open-ended survey comments, including comments from Extension leaders. 

#### 3.1.3. Need 3. Establish Connections with Medical Experts 

Individuals in our sample reported a high level of trust in official health organizations, including the CDC, physicians, and state and local health departments. As shown in Table 1, those who were less willing to administer vaccine education frequently cited lacking the medical expertise needed to do so and reported less trust in these sources. Enhancing connections and confidence with trusted health professionals may be important for programmatic success.

The estimation from a structural equation model revealed that trust in public health entities was positively correlated with comfort in addressing misinformation (r = 0.325, *p* < 0.001), willingness to speak out (r = 0.363, *p* < 0.001), and willingness to provide vaccine education (r = 0.576, *p* ≤ 0.001), as shown in Table 2. In addition, comfort for managing misinformation was associated with willingness to provide vaccine education (r = 0.496, *p* < 0.001), and comfort to manage misinformation was associated with willingness to speak out (r = 0.438, *p* < 0.001). Willingness to speak out was associated with willingness to provide vaccine education (r = 0.614, *p* < 0.001). Perceived severity of the disease was also associated with willingness to speak out (r = 257, *p* < 0.001), willingness to engage in vaccine education (r = 421, *p* < 0.001), trust in health professionals (r = 518, *p* < 0.001), and comfort in managing misinformation (r = 254, *p* < 0.001). The model fit was strong (CFI = 0.975, SRMR = 0.048, and RMSEA = 0.083). While the RMSEA estimate was above the threshold recommended by Hu and Bentler (1999), it was likely due to known properties of the statistic estimated with a low degrees of freedom model and not of great concern [63].

Overall, these findings highlight the importance of forging partnerships with health experts, findings that were later confirmed with our qualitative data. In particular, this supported the team’s hypothesis that a three-pronged approach of addressing discomfort with managing misinformation, Motivational Interviewing, addressing discomfort at managing interpersonal communication about difficult issues, and Neuromarketing Science-informed message design skills development to build trust with expert public health message providers would build confidence in vaccine education. The results displayed in Table 2 also offer some evidence for the important role that Neuromarketing Science can play in optimizing messaging about vaccine science and promoting vaccine adherence, addressing the complexity previously discussed about differences between a life-challenging vaccine pathogen and a preventative vaccine. Most of the self-report data underlying these results are from measures that conceptually tap emotional responses rather than cognitive processes. The strongest relationships reported in Table 2 involve the role of trust in driving attitudes and behavioral intentions. The complexity of vaccine science, along with the highly emotional communication environment surrounding vaccine adherence, likely makes varying levels of trust primarily emerge from emotional connections rather than from cold, cognitive evaluations. This seems to describe a specific, emotionally loaded environment for which Neuromarketing Science is particularly well equipped to help optimize messaging to promote vaccine adherence. 

#### 3.1.4. Need 4. Strengthen Science Media Literacy Skills to Counter Misinformation and Communicate Emerging Science 

Extension professionals often discussed pride in their abilities to integrate science with local knowledge. Nevertheless, a frequently noted barrier to promoting vaccine education was the perception that it is difficult to counter media misinformation. Extension professionals also discussed a need for understanding what sources provided enough information based on how quickly the science around COVID-19 evolved. Individuals need to know where to find enough good information and how to follow and communicate changing guidance. Professionals’ comfort addressing misinformation was positively correlated with science media literacy for source (r = 0.162, *p* < 0.001) and science media literacy for content (r = 0.132, *p* < 0.001), with levels of self-reports shown in Table 3. Thus, they can benefit from resources that strengthen media literacy skills, particularly for science information, and help to communicate emerging science and counter misinformation.

### 3.2. Qualitative Methods

#### 3.2.1. Focus Groups

The Zoom-based focus groups (*n* = 22) comprised six groups based on volunteers recruited from our needs assessment survey indicating they were not currently engaged in vaccine education. Participants were segmented based on willingness to participate in vaccine education (high, moderate, and low). The focus groups included 16 field educators or agents and 6 specialists, with 50% having 10 or more years of experience. They included 15 women, 6 men, and 1 nonbinary individual. 

#### 3.2.2. Focus Groups Results

Positively valenced focus group comments tended to emphasize gratitude for the opportunity to serve the community, expressing joy, fulfillment, empathy for constituents, and pride in expertise. Participants particularly expressed positive emotions related to Extension’s role to keep constituents informed of the latest research and to lead other efforts related to public health and safety, such as providing information about livestock vaccines for farmers. They noted that they feel pride and fulfillment in their community-driven outreach and education. The following quotes express these sentiments:

“I feel that if we want to be known for community wellness work, we should not sidestep an issue with such widespread impact on community wellness”.

“I understand vaccines and the science behind them. I educate my producers about Livestock vaccines all the time”.

“I would be willing to conduct vaccine education if it treats individuals who are afraid of vaccination or paranoid with respect and does not involve intimidation, humiliation, or even persuasion. Just share general information about vaccinations, how they work, and how they are developed”.

Negatively valenced focus group responses tended to emphasize how the pandemic had changed Extension roles, overwhelming individuals with the scope of their responsibilities. Enforcing COVID-19 restrictions and changing regulations led to resentment, frustration, and perceived loss of credibility. Many participants expressed that their constituents have already made up their minds about the COVID-19 vaccine. Personal beliefs about vaccine efficacy also affected attitudes toward providing this education. Several expressed the sentiment that they are not public health experts and vaccine promotion is outside of their expertise and job description. They also indicated that misinformation and the politicization of COVID-19 would make vaccine education difficult.

“I would like honest information that takes the emotional appeal out of this. People have picked sides and don’t seem willing to discuss holes in the message. Personal experiences of people around me do not match mainstream messaging. That creates fear and makes me slow to tell others what they should do”.

“The peer pressure of partisan politics within extension has made it difficult to continue pride in the work we do. Public health should have never become political, yet here we are”.

“I do not wish to be berated by members of the public as part of my job. I am a horticulturist, not a public health administrator”.

### 3.3. Expert Interviews

#### 3.3.1. Expert Interview Methods

Expert interviews (*n* = 10), 30 min in length, were conducted with Extension directors or administrators from each of the five Extension regions (two Western, two Southern, two Northeast, three North Central, and one 1890). They were recruited via contact with Regional Executive Directors. The interviews were semi-structured, open dialogue discussions between the experts and two of the researchers. They were conducted via Zoom, which recorded transcripts of the interviews. All of the data from the transcripts were anonymized to protect the identities of the participants. The resulting transcripts from these conversations were then evaluated using thematic analysis based on Braun and Clarke’s [64] methodology. The answers to the interview questions were coded by researchers and analyzed for shared themes across participants. 

#### 3.3.2. Expert Interview Results

The expert interviews produced a consensus that vaccine education aligned with Extension values. These include service, integrity, trust, credibility, excellence, respect, helping people, and providing information. As quoted by one Extension Director, “We can’t change who we are. We are still a science-based education organization”. Some, however, indicated that they had made the decision not to be involved in vaccine education because of the politicization of vaccines and pressure from powerful elected officials and stakeholders not supporting Extension vaccine education.

### 3.4. Neuromarketing Science

#### 3.4.1. Neuromarketing Science Methods

The Neuromarketing Science content testing involved delivering COVID-19 vaccine science education consistent with Extension values to test how Extension professionals cognitively and emotionally process and respond to content that frames COVID-19 vaccine education in the context of Extension values and provide insights from this test in the toolkit. 

Neuromarketing Science-based testing of responses to COVID-19 vaccine promotional messages (*n* = 51) was performed at the Conference of the Association of Extension Administrators, August 2022, in Orlando, FL, USA (*n* = 31), and at the conference of the National Association of Extension 4-H Youth Development Professionals, October 2022, in Madison, WI, USA (*n* = 20). Participants were recruited through Extension newsletters and on-site flyers. The majority of participants were women (66.7%). The average age of the participants was around 45 (M = 45.64, SD = 11.46), and 45.1% of the participants identified as White (*n* = 23), 52.9% identified as Black or African American (*n* = 27), 5.9% (*n* = 3) identified as Hispanic or Latino, and 5.9% (*n* = 3) identified as Native American or Alaskan Native. Participants came from all five Extension regions (2 Western, 26 Southern, 6 Northeast, 4 North Central, and 13 1890), and 43.1% (*n* = 22) had worked in the Extension for more than 10 years (43.1%, *n* = 22). They included 21 (41.2%) Extension field educators or agents, 8 (15.7%) Extension specialists, 5 (9.8%) Extension directors or administrators, and 17 (33.3%) Other. 

The testing stimuli included seven COVID-19 vaccine education message frames. Four reflected Extension professionals’ values [56], including value frames on education values, intellectual freedom, the role of Extension between the community and science, and commitment to humanity. Three of the stimuli were related to the emotional frames that emerged from the survey results and Zoom-based focus groups, including gratitude in serving the community, empathy for constituents, and pride in expertise. The emotional nature of formal Extension value statements as frames, especially when associated with the topic of vaccines, and the explicitly emotional frames based on specific emotions revealed through the survey are conceptually related to trust judgements. As noted earlier, trust is a core emotional judgement in the context of attitudes and behaviors related to vaccine science. Therefore, the stimuli that were tested through Neuromarketing Science emerged from the survey and focus group data in order to further improve the value and application of Neuromarketing Science in this project designed to ultimately lead to a communication toolkit for effective messaging promoting vaccine adherence in an environment filled with misinformation. Each Extension value frame was displayed on the computer screen for 20 s, and each emotional frame was displayed for 35 s. Each participant viewed all seven frames for the same amount of time and answered self-reported emotional and attitudinal questions on 1- to 11-point Likert scales after each frame. Skin conductance (physiological indicator of arousal), heart rate (indicator of cognitive resources allocation), and corrugator facial electromyography (negative emotional response) were recorded while participants read the message frames. 

#### 3.4.2. Neuromarketing Content Testing Results

This was a small sample study designed to gain qualitative insight, therefore data analysis and interpretation are only qualitatively directional and do not involve statistical significance testing. Neuromarketing Science-based measurements, as well as descriptive results from self-reported emotional and attitudinal questions, showed that the emotional frame regarding empathy for constituents performed the best in grabbing attention towards the message, as indicated by more cognitive resources allocated to encoding. Additionally, it elicited more willingness (M = 6.84, SD = 2.55) and comfort (M = 6.65, SD = 2.66) in vaccination education among Extension professionals while also minimizing negative emotions compared to the other two emotional frames. This was followed by the frame mentioning “pride in expertise”. In other words, Extension professionals value the importance of having and demonstrating empathy for community members and take pride in their specific expertise when engaging in educational programs.

Vaccine science content using “message framing” that resonates with specific values identified in the Extension professional’s creed are effective. Specifically, the statement “We believe that Extension is a link between the people and the ever-changing discoveries produced by expert scientists” elicited the greatest positive emotions (M = 6.53, SD = 2.67) while minimizing negative emotions (M = 2.35, SD = 2.37) among all four value frames. Additionally, participants showed more willingness (M = 6.96, SD = 2.57), felt more comfortable (M = 6.86, SD = 2.60), and were more competent (M = 6.47, SD = 2.60) towards vaccination education when exposed to this statement. However, the results also suggested that the frame using phrases such as “vaccine education is basic in stimulating individual initiative and self-determination” and “we have to make our lives and the work we do as Extension professionals useful to humanity” is not recommended for vaccine science education among Extension professionals, as it might trigger more negative emotions and perform poorly in driving willingness and comfort towards vaccination education. This result points to the practical usefulness of Neuromarketing Science in optimizing communication in a way that effectively avoids eliciting overly negative emotions and positively emotionally engaging the priority population for vaccine messaging. In summary, Extension professionals value being a critical link between “science” and “community members” who can benefit from science.

### 3.5. Pilot Workshops

#### 3.5.1. Pilot Workshop Methods

Based on needs assessment and Neuromarketing testing results, the WSU project team developed pilot workshops. Pilot workshops took place over three virtual sessions convened for EXCITE professionals as follows: Motivational Interviewing, 21 February 2023 (*n* = 40); Science Media Literacy, 28 February 2023 (*n* = 38); and Neuromarketing Science, 7 March 2023 (*n* = 37). Penultimate pilot workshops comprising three sessions convened virtually as follows: 6 June 2023, Motivational Interviewing (*n* = 86); Science Media Literacy, 7 June 2023 (*n* = 72); and Neuromarketing Science, 8 June 2023 (*n* = 60). Each session included PowerPoint slides narrated by project staff, discussion opportunities, and breakout practice in the first two sessions using worksheets drawn from the toolkit. Qualtrics surveys collected feedback at each session.

#### 3.5.2. Pilot Workshop Results

The participants’ responses across various workshops indicated varying levels of confidence gained in Motivational Interviewing, Science Media Literacy, and Neuromarketing skills for addressing vaccine hesitancy. 

#### 3.5.3. Key Strengths Identified in the Pilot Workshops

Participants were receptive to the material and engaged in the breakout sessions. Overall feedback was positive, and participants appeared to benefit from the material. The results reinforced the findings of the needs assessment and confirmed the direction of the concept testing. Responses improved from the initial pilot tests of the toolkit concepts to the tests of the complete toolkit in June. An example of participant feedback was as follows: “My biggest takeaway was the importance of engaging others and building collaborative relationships with those with subject matter expertise to help engage with others regarding matters of scientific communication”.

#### 3.5.4. Key Challenges and Opportunities Identified in the Workshops

Some participants indicated a need for more opportunities to practice skills earlier in the sessions. Participants also suggested including an opportunity to have a conversation with a community member. Participants appreciated the examples provided and expressed enthusiasm for new material and information. Some information such as understanding news coverage of a meta-analysis for Science Media Literacy seemed too complex to grasp comfortably in a single practice session and would require further deconstruction and simpler application to provide any practical relevance. The primary barrier for the pilot workshops was the attrition of participants when breakout sessions occurred. We therefore incorporated more comprehensive group activities and engaging interactive content into the main sessions, such as Zoom chat waterfalls, thereby fostering a more unified learning environment. This appeared to promote sustained engagement and retention. Rehearsal and timing checks also improved flow.

### 3.6. Integrated Findings from the Survey, Focus Groups, In-Depth Interviews, Neuromarketing Content Testing, and Pilot Testing

#### 3.6.1. The Integrated Model of Sustainable Health Decision-Making

Using the comprehensive needs assessment results, the WSU project team has developed the Integrated Model of Sustainable Health Decision-Making (see Figure 3), which has integrated Science Media Literacy (SML), Neuromarketing Science (NMS), and Motivational Interviewing (MI) to illustrate the mechanisms of community health behavior change. Within the Integrated Model of Sustainable Health Decision-Making, the Extension professionals engage in (1) synchronous and asynchronous professional development experiences focused on (2) developing and maintaining the role of trusted partners and messengers of health information, thus leveraging the positive alliance with priority populations to increase their motivation to engage in vaccinations by (3) providing priority populations with strategies to increase Science Media Literacy and increase their confidence in independently identifying trusted sources of health information and designing and disseminating marketing and social media content to priority populations. Neuromarketing Science most directly advances number 3 in the Integrated Model of Sustainable Health Decision-Making. However, as an approach that focuses on how the human brain processes, evaluates, and responds to health information, Neuromarketing Science enhances the effectiveness of each part of this model in promoting science-based health decision-making—in this case, vaccine adherence.

Figure 3 portrays four main nodes of action inherent in this theoretical framework. The Professional Development Core is comprised of the integration of Science Media Literacy Skills, Neuromarketing Science, and Motivational Interviewing Skills designed to support trust and collaboration between priority populations and trusted partners. Trusted partners encompass professionals who are equipped to provide credible health-related information and support priority populations’ vaccine uptake. Priority populations include individuals identified as having inequitable access to credible health information and services and/or having low rates of vaccine uptake. Within the Integrated Model of Sustainable Health Decision-Making, Extension professionals can locate themselves in all the nodes as trusted partners in their communities, members of priority populations, and facilitators of professional development. Additionally, they often create or cocreate trustworthy communication activities that educate and motivate community members and healthcare partners for sustainable improvements to well-being. Their key role, given their special areas of expertise, is to facilitate connections among each of these entities/activities in a co-collaborative manner to reduce vaccine hesitancy and increase vaccine uptake among priority populations. 

#### 3.6.2. Getting to the Heart and Mind of the Matter Toolkit 

The results of the comprehensive needs assessment, pilot workshops, and the Integrated Model of Sustainable Health Decision-Making informed the development of the innovative Getting to the Heart and Mind of the Matter toolkit (GHMM) [65]. The aim of GHMM was to provide Extension professionals with evidence-based professional development resources to equip and empower them to implement vaccine education in priority populations in their communities during the height of the COVID-19 pandemic and for ongoing success. Furthermore, the Integrated Model of Sustainable Health Decision-Making (see Figure 3) was the theoretical basis for the toolkit creation. GHMM “replace[d] Extension’s use of the ‘expert model’ (with the answers going into the community to solve problems) with a community-informed approach with Extension as a trusted messenger and partner in working alongside community members and community organizations. This represents a shift that has been occurring over time across the Extension system, but which is routinely incorporated into the model of the EXCITE work” [66]. 

The GHMM toolkit draws on a three-pronged approach to health education: SML, MI, and NMS to provide Extension professionals with skills to engage, educate, and empower priority populations about vaccinations by (1) shifting the emotional valence of these conversations from negative (e.g., anxiety, fear, and uncertainty) to positive (e.g., confidence, personal agency, and positive expectations); (2) providing Extension professionals with the tools to support priority populations’ use of evidence-based decision-making regarding vaccinations; and (3) increasing the use of brain-friendly (see the previous section on Neuromarketing Science) vaccine promotion materials among priority populations. This specific role of Neuromarketing Science for content optimization is intended to arm communicators with the confidence and feelings of trust crucial for making them effective communicators in delivering brain-friendly messaging to promote vaccine adherence.

GHMM is founded on a strong evidence base for reducing vaccine hesitancy and increasing health behaviors among priority populations. SML, MI, and NMS focus on increasing positive emotions (e.g., hope, confidence, and commitment) and reducing negative emotions (e.g., fear, anxiety, and indecision). Vaccine-related scholarships have long recognized the importance of emotions in vaccine behaviors [67]. The COVID-19 pandemic has centered on the role of emotions in influencing vaccine behavior [68,69], with recent research focusing on how positive emotions during social interactions could support vaccine uptake behaviors [70]. The application of Neuromarketing Science highlights a unique body of knowledge for developing brain-friendly vaccine science messaging and the potential to include rigorous content testing in developing, designing, and delivering vaccine promotion content that is brain-friendly and optimally effective at reducing psychological barriers to vaccine uptake. 

GHMM consists of didactic content, self-paced activities, and video examples. The first section of GHMM introduces Motivational Interviewing (MI) as the foundation of how we communicate with individuals to help them feel understood, confident, and in control to make good decisions. The second section on Science Media Literacy (SML) focuses on how our SML confidence and skills help us access, analyze, evaluate, create, and act using media in various forms. The third section, about Neuromarketing Science (NM), discusses how to apply brain-friendly messaging to develop vaccine education content. This section gives health communicators a basic understanding of Neuromarketing, the science of how the human brain processes information, along with specific practical tips for designing and delivering messaging that promotes a positive emotional response that facilitates beneficial cognitive processing of information. To date, GHMM has been used by 33 EXCITE teams across the U.S. to provide health communication information to priority populations and engage Extension professionals in continued professional development to use GHMM content and tools [71]. The project is just beginning to assess the effectiveness of the GHMM toolkit in enhancing professionals’ willingness and confidence in vaccine education. 

## 4. Discussion

Our team performed a comprehensive needs assessment to inform the development of a toolkit to provide Extension professionals with evidence-based professional development resources. These tools equipped and empowered them to implement vaccine education for priority populations in their communities during the height of the COVID-19 pandemic and provide a solution to address Extension professionals’ potential hesitancy [12]. The needs assessment, which involved a mixed methods survey, follow-up focus groups, in-depth interviews, and Neuromarketing Science content testing methods, identified four essential needs: tailoring training based on Extension roles, prioritizing preserving community trust and professional credibility, establishing connections with medical experts, and strengthening Science Media Literacy skills to counter misinformation and communicate emerging science.

Overall, Extension professionals value their positions’ autonomy, adapting to their communities’ needs with evidence-based solutions, which builds rapport with local leaders, small businesses, and families. They enjoy the bottom-up approach by which their communities guide their priorities, and they do not perceive that they are mandated to provide one-size-fits-all programming. The GHMM design allows training tailored to community needs and connections with medical experts. By also taking an integrated approach, the GHMM toolkit allows Extension professionals to enhance their communication skills and allows them to develop and maintain trust within their communities [65].

The social credit of professionals in health outreach programming in their communities was challenging to maintain and build during the pandemic. Many hesitated to spend it on something as politically charged and controversial as the COVID-19 vaccine. The GHMM content and tools were designed so that professionals could preserve community trust as suppliers of credible information in an apolitical and non-confrontational way, which was essential given the challenges educators face when promoting immunization education [4,5,6,72,73].

Extension professionals have already communicated updated research information to their local communities in various innovative ways. These strategies demonstrate that they already possess an established Science Media Literacy skillset that the GHMM toolkit can strengthen for building confidence and sharing for vaccine education in priority populations. The success of the toolkit and training provides evidence for how GHMM’s integrative process can build on established media literacy-oriented intervention techniques for developing resistance to misinformation [26,27,28,60].

This study sought to understand Extension’s role in rural health education but was limited in a few ways. First, although the expert interviews were meant to offer insights from leadership from different areas, it is not representative of all of Extension’s leadership, and some meaningful contributions could have been added if more participants were included. Additionally, the results presented in this study are specific to Extension educators and may not fully translate to other contexts or to other groups of vaccine educators. However, given the broad reach of Extension facilitators in the U.S. to rural communities, the research presented here shows promise for developing tailored resources to promote vaccines within this context, consistent with the purpose of the toolkit. In addition, the findings from Neuromarketing, while small in sample size, included representations across regions and demographic groups. The fact that the results are based on established theoretical principles also suggests that the findings of this study have potential for future testing of this process framework in other settings and with other samples, including in international contexts.

Extension professionals are willing to work hard for the communities they care about, to educate when they see a need, and to help in whatever way their position allows. The toolkit’s needs assessment and trial run through Zoom workshops with Extension professionals demonstrated the importance of an iterative design that can take advantage of regional input and flexibility in the application of data collection and skill development tools. These workshops provided a unique opportunity for partnership on the data collection and analysis throughout the project, strengthening its overall effectiveness. For example, some of the measures in the survey were refined based on discussion groups held prior to data collection with professionals during one of their regular national planning meetings. As we learned how busy they were and observed their range of responsibilities, we realized that the order of the workshops needed to be adaptable and customizable, without reliance on sustained attendance or having read the toolkit beforehand. We also learned in meetings and workshop pilots that breakout sessions resulted in a loss of participants, reinforcing the need to design short, engaging workshops for the final YouTube versions. Finally, we learned the importance of word choice, jargon, and vernacular, in that we were outsiders who needed to learn the culture of Extension and the communities they serve.

The needs assessment findings address evidence-based strategies for promoting vaccine education while addressing the unique needs of Extension professionals to maintain trust and credibility within the community. Feedback from the pilot tests was overwhelmingly positive, with only minor changes required to design the toolkit and workshops. Feedback at the workshop presentations led to final updates and the penultimate toolkit delivered to EXCITE in July 2023. Minor updates have continued throughout 2023–2024 based on continuing feedback sought by the team and updates to the information about science and the media landscape.

## 5. Conclusions

Because Cooperative Extension professionals live and work within their communities, they are trusted messengers for providing research-based information and working with trusted partners and community members to address local needs. Maintaining their established trust as community educators is critical when addressing controversial topics such as the COVID-19 vaccine. The strategies, packaged within the toolkit entitled *Getting to the Heart and Mind of the Matter: A Toolkit and Workshops for Building Confidence in Being a Trusted Messenger of Health Information,* underwent an iterative process with Extension professionals to ensure its viable use for health education. 

Extension professionals now have a toolkit package to use in synchronous and asynchronous professional development experiences with priority populations and to help them effectively design and disseminate marketing and social media content within their communities. They can now provide strategies to increase Science Media Literacy through increased confidence in independently identifying trusted sources of health information. Moreover, Extension professionals can maintain their role as trusted partners and messengers of information, increasing motivation towards vaccination and building healthier communities.

## Figures and Tables

**Figure 1 vaccines-12-00869-f001:**
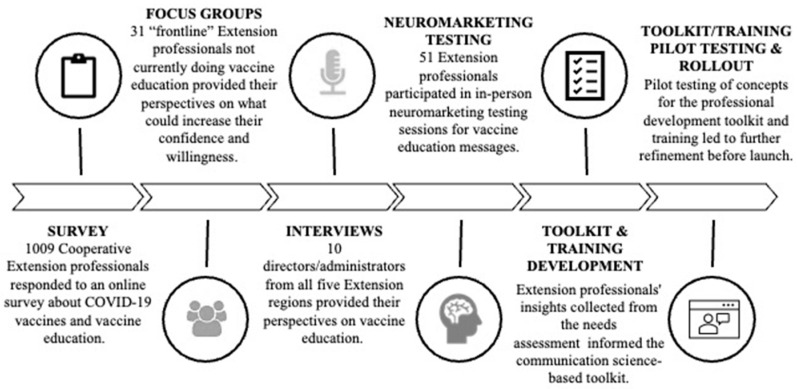
Inclusive approach capturing insights on vaccine education trust and willingness across the Extension system.

**Figure 2 vaccines-12-00869-f002:**
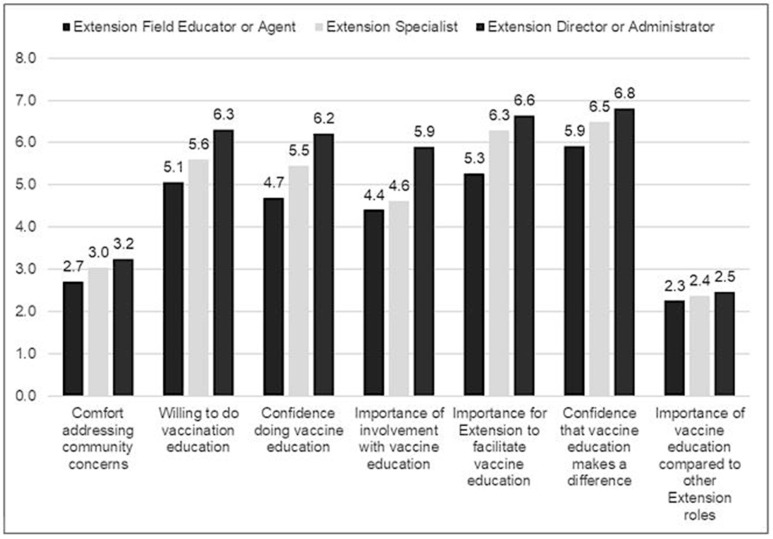
Vaccine education programming responses by roles.

**Figure 3 vaccines-12-00869-f003:**
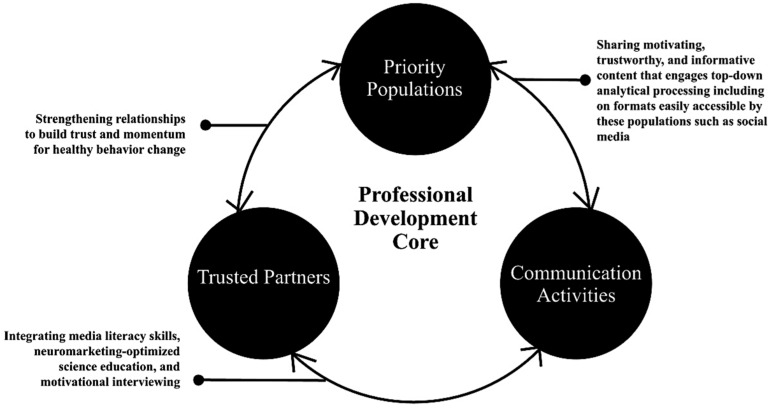
Integrated Model of Sustainable Health Decision-Making.

**Table 1 vaccines-12-00869-t001:** Measures comprising the constructs in the structural model.

Measures	*Mean*	*St. D*	*n*	Range
**Willingness to Do Vaccine Education (*alpha* ** **= 0.857)**				10-point Likert Scale (1 = Not at all to 10 = Very).
On a scale of 1 to 10, how willing are you to do vaccination education?	5.38	3.28	886	
On a scale of 1 to 10, how confident are you doing vaccine education?	5.17	3.15	852	
**Efficacy and Trust for Vaccine Education (*alpha* ** **= 0.89)**				4-point Likert Scale (1 = Strongly disagree to 4 = Strongly agree or 1 = Not at all to 4 = Very much).
I believe the COVID-19 vaccine will be effective in preventing the coronavirus disease.	2.93	0.980	813	
How likely are you to recommend getting the COVID-19 vaccine to others?	2.49	0.778	819	
I believe the COVID-19 vaccine works in preventing the coronavirus disease.	2.75	0.953	815	
How much do you trust the public health agencies that recommend you get a COVID-19 vaccine?	3.15	1.04	841	
**Willingness to Speak Out (*alpha* = 0.915)** If the topic of COVID-19 vaccines came up in these settings, how willing would you be to join in the conversation?				4-point Likert Scale (1 = Very unwilling to 4 = Very willing).
At a community meeting	2.74	0.919	752	
With Extension coworkers	3.11	0.841	750	
With Extension funders or stakeholders	2.68	0.968	749	
With Extension constituents/clients	2.67	0.981	749	
**Comfort in Addressing Misinformation**				5-point Likert Scale (1 = Very Uncomfortable to 5 = Very comfortable.
How comfortable do you feel addressing misinformation about the COVID-19 vaccine?	2.86	1.25	839	

Note: For constructs employed, high scores are considered positive responses to the item statement, while a low score indicates a negative response.

**Table 2 vaccines-12-00869-t002:** Estimated correlations for the latent variables.

Variables	1	2	3	4	5	6	7
1. Willingness to do Vaccine Education	-						
2. Willingness to Speak-Out	0.614 **	-					
3. Science Media Literacy for Source	−0.008	−0.011	-				
4. Science Media literacy for Content	0.006	0.016	0.934	-			
5.Perceived Severity of COVID-19	0.421 **	0.257 **	0.054	0.0350	-		
6. Trust in Public Health Agencies	0.576 **	0.363 **	−0.092	−0.099	0.518 **	-	
7. Willingness to Address Misinformation	0.496 **	0.438 **	0.162 **	0.132 **	0.254 **	0.325 **	-

** Indicates *p* < 0.01.

**Table 3 vaccines-12-00869-t003:** Science Media Literacy for “sources” and “content” (alpha = 0.95).

Measures	*Mean*	*St. D*	*n*
**Science Media Literacy for Sources (*alpha* = 0.91)**			
I check whether those who create science news know about the topic.	3.72	1.33	8.13
I think about what point of view a science broadcaster or writer is trying to support.	4.12	1.27	811
I look to see if those who share science news on social media have checked the accuracy of their facts.	3.65	1.41	794
I think about whether sources of science news have my best interests in mind.	4.00	1.33	807
I think about whether those who provide science information might be doing so to gain power or profit.	3.94	1.41	812
I get science news from multiple sources to make sure I get the full story.	4.20	1.21	809
**Science Media Literacy for Content (*alpha* = 0.90)**			
I think about how scientists can draw different conclusions from the same science facts.	3.97	1.20	787
I check to see if a science fact in a news story is backed up by a credible source.	4.10	1.21	784
I check to see if a picture or graph accurately matches the scientific information it represents.	3.86	1.27	785
I check to see if the science news I read is up to date.	4.23	1.20	777
I think about whether a news story with real science facts could still lead to a false conclusion.	3.82	1.20	782
I have changed my thinking about a science topic when I received new information.	3.62	1.04	782

Note: Constructs were measured using a 6-point scale, from 1 = “never” to 6 = “all the time”.

## Data Availability

The data presented in this study are available on request from the corresponding author.

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
