# Peer review of "Integrating Science Media Literacy, Motivational Interviewing, and Neuromarketing Science to Increase Vaccine Education Confidence among U.S. Extension Professionals"

_vaccines, 2024, doi:10.3390/vaccines12080869_

Round 1

Reviewer 1 Report

Comments and Suggestions for Authors

I read the article by Erica Weintraub Austin et al. entitled "Integrating Science Media Literacy, Motivational Interviewing and Neuromarketing Science to Increase Vaccine Education Confidence Among U.S." Extension Professionals'. It is an interesting article which "presents an Integrative Model of Sustainable Health Decision Making and a toolkit to equip U.S. Extension professionals with knowledge and skills to engage in adult immunization education. The objective was to reduce mistrust and increase willingness and confidence towards delivering vaccination education.”

Here are my observations:

 Introduction

- The introduction is too long, I would suggest to limit it, while you should increase the references

-At the end of the introduction, please specify the exact objectives of the project but mainly of this article.

Materials and Methods

-Despite the much information given by the authors about the sample, it is not clear whether this sample is representative (over-representation of women does not support representativeness) , please the authors clarify whether they checked the representativeness of the sample.

-For the interviews that were conducted, what methodology did you use to analyse them?

-What statistical tests did you use?

Results

The results are missing statistical controls, and statistical tables, please add them. The authors in several sections draw conclusions without statistical justification.

Discussion

Can the results of the article be generalized to other populations?

Add limitations at the end of the discussion.

Line 721-722: "Appendix A Getting to the Heart and Mind of the Matter Toolkit: WSU resources" delete it.

Author Response

Thank you for your suggestions, which have been quite helpful to us. Our responses follow.

  • Comment 1: The introduction is too long, I would suggest to limit it, while you should increase the references; At the end of the introduction, please specify the exact objectives of the project but mainly of this article.
  • Response 1: 
    • Thank you for your suggestions to shorten and update the introduction. We have shortened and updated the introduction to increase clarity, conciseness, and to ensure that our references are current and pertinent to the manuscript. Additionally, we aligned the in-text references with our reference list.
    • Lines 86-93, additional material regarding emotion and international contexts, as follows: “Similar results are present in the international arena [16]. For example, the results from a survey conducted in 126 countries indicated that in countries with high levels of trust in science, individuals were more vaccine confident [17]. Similarly, the results of a large-scale survey in Asian and Western countries indicated that trust in government decreases the impact of misinformation on vaccine uptake [18]. This points to the complicated affective environment of vaccine science education. Traditional cognitive processes such as knowledge take a back seat to affective processes driven by emotion and affective cognitions such as trust, willingness, expectancies, and efficacy.”
    • Lines 97-104, additional justification of the sample and objectives of the study: However, the Cooperative Extension professionals are well-suited to deliver impactful vaccine education since they can leverage community partnerships and have built trusting relationships with community members through other Extension initiatives (e.g., agricultural education, nutrition, youth development) – characteristics that can support vaccine education initiatives [19]. Below we describe the elements of the needs assessment to further understand the Cooperative Extension professionals’ perspectives and develop a set of educational materials that could increase their confidence in delivering vaccine education to priority populations.”
    • Lines 138-158, the Science Media Literacy section has been made more concise, as follows: “Within the context our EXCITE needs assessment, the project team hypothesized that equipping Extension professionals with MI skills would equip them with tools to (1) reduce negativity response spiral and optimize resonance with COVID-19 vaccine-related education and outreach, thus (2) increasing their willingness and confidence in engaging in COVID-19 vaccine education with priority populations. The project team also hypothesized that the use of MI by Extension professionals to strengthen trust and positive relationships with priority populations would reduce the boomerang effect among priority populations – or rejection of information contrary to one’s beliefs [46] -- and support these populations in using science media literacy skills to seek out credible sources of information [47].”
  • Comment 2: Despite the much information given by the authors about the sample, it is not clear whether this sample is representative (over-representation of women does not support representativeness) , please the authors clarify whether they checked the representativeness of the sample.”
  • Response 2: 
    • We have now added the following comments to discuss representation. “Our team aimed to reach a sample of Extension professionals that would be representative of the organization, including individuals from various geographic areas and Extension roles. “
    • Lines 282-283: Justification for the high proportion of females in the sample: “Additionally, our sample has a higher proportion of females, reflecting the organization's overall employee sex distribution [59]” Note: This citation states that women faculty membership in the American Association for Agricultural Education, which is a primary organization for Cooperative Extension faculty, was 14.6% in 2003 and had improved only to 21.9% in 2019. Citation: Cline, L. L., Rosson, H., & Weeks, P. P. (2019). Women Faculty in Postsecondary Agricultural and Extension Education: A Fifteen Year Update. Journal of Agricultural Education, 60(2), 1–14. https://doi.org/10.5032/jae.2019.02001
  • Comment 3: For the interviews that were conducted, what methodology did you use to analyse them?
  • Response 3: 
    • Lines 429-435, regarding the analysis procedures for the qualitative data collected:  “The interviews were semi-structured, open dialogue discussions between the experts and two of the researchers.  They were conducted via Zoom, which recorded transcripts of the interviews. All of the data from the transcripts were anonymized to protect the identities of participants. The resulting transcripts from these conversations were then evaluated using thematic analysis based on Braun and Clarke’s [64] methodology.  The answers to the interview questions were coded by researchers and analyzed for shared themes across participants. “
  • Comment 4: What statistical tests did you use? … The results are missing statistical controls, and statistical tables, please add them…. The authors in several sections draw conclusions without statistical justification.”
  • Response 4: 
    • In addition to descriptive statistics, our team used a structural equation model to evaluate the relationships between trust in public health entities, comfort in addressing misinformation, and willingness to engage in vaccine education and speak out. We have now clarified information about this analytical strategy in the manuscript including our use of controls in the model. We also now include a correlation table.
    • Beginning Line 331: We include significance testing for relationships among the latent constructs. We have added a correlation table as requested.
    • Lines 478-481, we explain why significance testing was not used for the Neurotesting content analysis: “This was a small sample study designed to gain qualitative insight so data analysis and interpretation is only qualitative directional and does not involve statistical significance testing.”
  • Comment 5: Can the results of the article be generalized to other populations?... Add limitations at the end of the discussion.
  • Response 5: 
    • Thank you for your comments and suggestions. To address the issue of generalizability, we have added a section detailing the analysis and methods of the interviews, along with a reference. This provides a clearer context for our findings. We have also included a section at the end of the discussion where we outline the study's limitations. We now include the following at Lines 666-680: “This study sought to understand Extension’s role in rural health education but was limited in a few ways. First, although the expert interviews were meant to offer insights from leadership from different areas, it is not representative of all of Extension’s leadership and some meaningful contributions could have been added if more participants were included.  Additionally, the results presented in this study are specific to Extension educators and may not fully translate to other contexts or to other groups of vaccine educators. However, given the broad reach of Extension facilitators in the US to rural communities, the research presented here shows promise for developing tailored resources to promote vaccines within this context, consistent with the purpose of the toolkit. In addition, the findings from neurotesting, while small in sample size, included representation across regions and demographic groups. The fact that the results are based on established theoretical principles also suggests that the findings of this study has potential for future testing of this process framework in other settings and with other samples, including in international contexts.”
  • Comment 6: Line 721-722: "Appendix A Getting to the Heart and Mind of the Matter Toolkit: WSU resources" delete it.
  • Response 6: 
    • This has been deleted as requested.

Reviewer 2 Report

Comments and Suggestions for Authors

The article refers to the exciting role of neuromarketing in vaccine adherence. The rationale is adequate in analyzing the specific group of health professionals. The general concern may be that the number of individuals interviewed does not provide solid statistical significance, specifically in the commitment assessment of directors. The directors may orally commit, but there is still some resistance to how they foresee the role of the vaccine in the disease, according to the data. Neuromarketing may aid the process, but no objective analysis allows the authors to demonstrate its impact on decision-making; a validated questionnaire should be used.

It would be interesting to monitor the opinions on a longitudinal study. In COVID-19, the experience of prior vaccination, after vaccination, and nowadays may be crucial. Does the decrease in patient death significantly affect the neuromarketing of vaccines?

The authors should also focus on the data published in other countries to compare with US data. Several interesting studies were not cited, eg doi: 10.1002/cl2.1352

Finally, the authors should correct how the references are cited in the text and at the end of the article.

Author Response

Thank you very much for your comments and questions, which have helped to strengthen our manuscript. Our responses to your inquiries follow.

Comment 1: Neuromarketing may aid the process, but no objective analysis allows the authors to demonstrate its impact on decision-making; a validated questionnaire should be used.

It would be interesting to monitor the opinions on a longitudinal study. In COVID-19, the experience of prior vaccination, after vaccination, and nowadays may be crucial. Does the decrease in patient death significantly affect the neuromarketing of vaccines?

Response 1: This study focuses on understanding neuromarketing content principles for vaccine health promotion decisionmaking, and this study is a formative, qualitative study for precampaign planning. A comprehensive longitudinal study would require a different design. To address the reviewer’s concerns and questions, the explanation of Neuromarketing has been expanded as follows, beginning on Line 147:

“Neuromarketing Science refers to the systematic study of how the human mind – embodied in the nervous system – processes and responds to media content and technology [48]. This field consists of two related yet independent specialized areas. Neuromarketing Science consists of a body of knowledge about how the human mind – embodied in the brain – works and processes lived experiences like being exposed to vaccine education content. It also contains a body of formal measures and methods for conducting primary research. The field of Neuromarketing Science serves as an element of the needs assessment. Both areas of Neuromarketing Science were applied to this specific project.

The value of Neuromarketing Science, as a field and conceptual foundation for this project is that it provides actionable insights to increase the effectiveness of content including health communication, advertising, news, entertainment, and politics spanning communication and media channels [49]. Lee et al. [50] recently outlined how this approach can inform strategic communication to engage organizational stakeholders such as Extension professionals and community members through what is termed in this project, brain friendly content.

The application of Neuromarketing Science to the needs assessment for this project was two pronged. The needs assessment involved conducting qualitative/interpretive primary research using Neuromarketing measures and methods. This consisted of a designed experiment testing the impact of vaccine visual cues and message framing on responses to Covid-19 vaccine messaging. This was a small sample study grounded in qualitative directional interpretation of data rather than statistical significance testing. Valid techniques for data analysis in Neuromarketing Science encompass projects using statistical significance testing and projects only using qualitative directional data analysis and interpretation [51].

The application of Neuromarketing Science in this project also focused on developing content for the toolkit grounded in the accumulated body of existing Neuromarketing knowledge focused on how the human mind processes and responds to media content and technology. This is a body of knowledge that has been growing since the 1980s [52]. This body of knowledge can be used to produce practical insights for communication professionals that guide the production of what is termed brain friendly content – content optimized for how the human mind processes media. The practical insights that emerge from Neuromarketing Science are rooted in formal principles.”

Comment 2: The authors should also focus on the data published in other countries to compare with US data. Several interesting studies were not cited, eg doi: 10.1002/cl2.1352. Finally, the authors should correct how the references are cited in the text and at the end of the article.

Response 2: Thank you for your suggestion. We have addressed your comment by including information from other countries. We have also included information from the suggested article. We also have checked how the references are cited in the text and at the end of the article to correct mistakes that were made previously. As already noted, we have added references throughout the introduction and methods as appropriate to provide evidence for our assertions.

Comment 3: Reviewer 2 indicated that the explanation of the methods could be improved.

Response 3: We have included additional information about the methods as follows.

    • Lines 252-261, Additional explanation of sources for survey measures: “Questions were adapted from previous Extension assessment tools, CDC rapid community assessment tools, previously published studies by the team, and science media literacy measures adapted from Austin et al. [24] for this study. The survey also included open-ended questions that asked about what would positively impact respondents’ willingness and confidence to engage in COVID-19 vaccine education as well as op-portunities to provide any other comments. At the end of the survey, respondents were asked to provide their email address if they were willing to participate in a focus group. A WSU Extension Educator led survey design based on recent Extension surveys, established measurement constructs, communication theory and the Total Design Method [58]. “
    • Lines 283-289, Additional explanation of measures newly developed for this survey: “Science media literacy, a newly created construct for this study, was measured using an updated version of Austin et al.’s [24] construct to better emphasize science information versus the scientific method as they had recommended, consistent with McClune and Jarman’s [61] five categories of knowledge and skills for critically consuming science news, the National Association for Media Literacy Education [15] core principles, and Bergstrom et al.’s [62] recommended skills for understanding science as a competent outsider.”
    • We have added a section detailing the analysis and methods of the interviews, along with a reference. This provides a clearer context for our findings. We have also included a section at the end of the discussion where we outline the study's limitations. We now include the following at Lines 666-680: “This study sought to understand Extension’s role in rural health education but was limited in a few ways. First, although the expert interviews were meant to offer insights from leadership from different areas, it is not representative of all of Extension’s leadership and some meaningful contributions could have been added if more participants were included.  Additionally, the results presented in this study are specific to Extension educators and may not fully translate to other contexts or to other groups of vaccine educators. However, given the broad reach of Extension facilitators in the US to rural communities, the research presented here shows promise for developing tailored resources to promote vaccines within this context, consistent with the purpose of the toolkit. In addition, the findings from neurotesting, while small in sample size, included representation across regions and demographic groups. The fact that the results are based on established theoretical principles also suggests that the findings of this study has potential for future testing of this process framework in other settings and with other samples, including in international contexts.”
    • We included this information about the analysis procedures for the qualitative data collected:  “The interviews were semi-structured, open dialogue discussions between the experts and two of the researchers.  They were conducted via Zoom, which recorded transcripts of the interviews. All of the data from the transcripts were anonymized to protect the identities of participants. The resulting transcripts from these conversations were then evaluated using thematic analysis based on Braun and Clarke’s [64] methodology.  The answers to the interview questions were coded by researchers and analyzed for shared themes across participants. “ 
    • We explain that "our team aimed to reach a sample of Extension professionals that would be representative of the organization, including individuals from various geographic areas and Extension roles." and we included this additional information to justify the sample, in Lines 282-283 explaining the high proportion of females in the sample: “Additionally, our sample has a higher proportion of females, reflecting the organization's overall employee sex distribution [59]” Note: This citation states that women faculty membership in the American Association for Agricultural Education, which is a primary organization for Cooperative Extension faculty, was 14.6% in 2003 and had improved only to 21.9% in 2019. Citation: Cline, L. L., Rosson, H., & Weeks, P. P. (2019). Women Faculty in Postsecondary Agricultural and Extension Education: A Fifteen Year Update. Journal of Agricultural Education, 60(2), 1–14. https://doi.org/10.5032/jae.2019.02001

Round 2

Reviewer 2 Report

Comments and Suggestions for Authors

The authors have tried to respond to the queries. Unfortunately, the responses have generated more questions regarding the impact of the possible study. Table 2  is essential; however, there is reasonable doubt about the effects of neuromarketing in vaccine therapy. The doubt concerns the difference between a life-challenging vaccine pathogen and a routine vaccine used as preventive medicine. I am still not convinced of the role of neuromarketing in vaccine adherence.

Author Response

Thank you for inviting us to provide additional refinements to our manuscript and for the helpful suggestions that have aided our revision process. We have described each of the updates we have provided in response to Reviewer 2’s suggestions as follows.

Comment 1: “The authors have tried to respond to the queries. Unfortunately, the responses have generated more questions regarding the impact of the possible study. Table 2  is essential; however, there is reasonable doubt about the effects of neuromarketing in vaccine therapy. The doubt concerns the difference between a life-challenging vaccine pathogen and a routine vaccine used as preventive medicine. I am still not convinced of the role of neuromarketing in vaccine adherence.”

Response 1: To explain the role of neuromarketing in vaccine therapy, we have made slight edits beginning on line 160, as follows:

This body of knowledge, built on over 40 years of scientific research on mental processes underlying media effects, provides actionable insights into producing all forms of effective media content including vaccine science messaging in support of vaccine therapy. The field of Neuromarketing Science also contains a body of formal measures and methods for conducting primary research. Neuromarketing Science therefore served as a rigorous element of the needs assessment. Both areas of Neuromarketing Science were applied to this specific project.

We have provided additional content to explain the role of neuromarketing in vaccine therapy  beginning on line 167:

Neuromarketing Science has an indirect, all though crucial, role to play in vaccine therapy. The benefits of vaccine therapy are ideally realized through voluntary, in-formed health decisions by individuals on a mass scale. This requires extremely effective communication and messaging about vaccines and vaccine science in support of advancing vaccine therapy. This is especially challenging in the current highly emotional and politicized communication environment surrounding the topic of vaccine therapy, including but also reaching beyond COVID-19 vaccination. Neuromarketing Science therefore provides a scientifically rigorous and practically powerful approach to informing the production and delivery of critical communication content about the importance of vaccines and vaccine therapy.

The particular role that Neuromarketing Science plays in vaccine therapy is to optimize communication content strategies that can empower individuals to make science-based decisions concerning acceptance or rejection of vaccine therapy. Neuro-marketing is an approach to testing content production that can optimize messaging about vaccines by preventing a negative cascade of emotions that hinder effective health decision making. This makes possible a communication environment that facilitates cognitive processing of vaccine science information and supports informed health decision-making that distinguishing accurate scientific information from misinformation.

To address the particular concern regarding the difference between a life-challenging vaccine pathogen and a routine vaccine used as preventive medicine in the role of neuromarketing in vaccine adherence we have added the following, beginning on Line 185:

This process is how Neuromarketing Science can play an indirect role in vaccine therapy that directly supports healthy decision-making that is required in vaccine adherence. Informed health decision-making should be the goal of all messaging about vaccination and vaccine adherence. Facilitating informed decision-making in an emotionally fraught environment is especially crucial given the complexities of vaccine science that encompasses highly complex topics such as the difference between a life-challenging vaccine pathogen and a routine vaccine used as preventative medicine. This is what made Neuromarketing Science a particularly useful tool for the needs assessment to produce messaging most likely to be effective in the scientifically complex, highly emotional health communication environment faced by the EXCITE team.  

The value of Neuromarketing Science, as a field and conceptual foundation for this project, has been demonstrated repeatedly in its history of providing actionable insights to increase the effectiveness of content for health communication, advertising, news, entertainment, and politics spanning communication and media channels [49]. Lee et al. [50] recently outlined how this approach can inform strategic communication to engage organizational stakeholders such as Extension professionals and community members through what is termed in this project, brain friendly content.

Comment 2: The reviewer says the research design must be improved:

Response 2: In lines 264-272 we provided additional context regarding how Neuromarketing testing is performed.

This was followed by empirical applied Neuromarketing content testing of ways of framing the topic of COVID-19 vaccine education within specific Extension values. This was done using a portable Neuromarketing lab at two Extension professionals’ conferences, one of which attracted a particularly diverse set of participants. A strength of Neuromarketing methods is the ability to test how individuals unconsciously and consciously mentally process and respond to content. The research methods of Neuromarketing Science holistically combine physiological indicators of cognitive and emotional processes with self-report and behavioral measures to gain valid and practically valuable insights into how individuals mentally process and respond to content [51, 52].

In lines 531-538 we explain more particulars of the method as follows:

Each Extension value frame was displayed on the computer screen for 20 seconds, and each emotional frame was displayed for 35 seconds. Each participant viewed all seven frames for the same amount of time and answered self-reported emotional and attitudinal questions on 1 to 11 Likert scales after each frame. Skin conductance (physiological indicator of arousal), heart rate (indicator of cognitive resources allocation), and corrugator facial electromyography (negative emotional response) were recorded while participants read the message frames.

In addition, given the reviewer’s concern that we needed to provide more information about the research design, we included additional information about the Neuromarketing stimuli in Lines 522-531 as follows:

The emotional nature of formal Extension value statements as frames, especially when associated with the topic of vaccines, and the explicitly emotional frames based on specific emotions revealed through the survey, are conceptually related to trust judgements. As noted earlier, trust is a core emotional judgement in the context of attitudes and behaviors related to vaccine science. Therefore, the stimuli that were tested through Neuromarketing Science emerged from the survey and focus group data in order to further improve the value and application of Neuromarketing Science in this project designed to ultimately lead to a communication toolkit for effective messaging promoting vaccine adherence in an environment filled with misinformation.

Comment 3: The reviewer indicated that the explanation of research results needed to be improved, particularly with respect to the role of neuromarketing.

Response 3: We therefore added material such as this in Lines 338-343:

Extension leaders and those who worked in Family and Consumer Sciences and Community Development expressed the highest levels of willingness and confidence.   These Extension professionals may have more experience with vaccine education which could shape their responses to the survey, as well as how their brains process the specific content that was tested later through Neuromarketing.

Comment 4: Table 2  is essential; however, there is reasonable doubt about the effects of neuromarketing in vaccine therapy.

Response 4: Thank you for your comments regarding the value of Table 2. To add additional value, we added additional explanation given the need expressed for more explanation of research results, as follows, in Lines 394-406:

The results displayed in Table 2 also offer some evidence for the important role that Neuromarketing Science can play in optimizing messaging about vaccine science and promoting vaccine adherence – addressing the complexity previously discussed about differences between a life challenging vaccine pathogen and a preventative vaccine. Most of the self-report data underlying these results are from measures that conceptually tap emotional responses rather than cognitive processes. The strongest relationships reported in Table 2 involve the role of trust in driving attitudes and behavioral intentions. The complexity of vaccine science, along with the highly emotional communication environment surrounding vaccine adherence, likely makes varying levels of trust primarily emerge from emotional connections rather than from cold, cognitive evaluations. This seems to describe a specific, emotionally loaded, environment for which Neuromarketing Science is particularly well-equipped to help optimize messaging to promote vaccine adherence. 

Additional implications of related to Neuromarketing for the results has been added in Lines 560-570:

However, the results also suggested that the frame using phrases such as “vaccine education is basic in stimulating individual initiative and self-determination" and “we have to make our lives and the work we do as Extension professionals useful to humanity” is not recommended for vaccine science education among Extension professionals, as it might trigger more negative emotions and perform poorly in driving willingness and comfort towards vaccination education. This result points to the practical usefulness of Neuromarketing Science in optimizing communication in a way that effectively avoids eliciting overly negative emotions and positively emotionally engaging the priority population for vaccine messaging. In summary, Extension professionals value being a critical link between “Science” and “Community Members” who can benefit from science.

How Neuromarketing specifically informs the theoretical model has been added in Lines 627-632, as follows:

Neuromarketing Science most directly advances number 3 in the Integrated Model of Sustainable Health Decision-Making. However, as an approach that focuses on how the human brain processes, evaluates, and responds to health information, Neuromarketing Science enhances the effectiveness of each part of this model in promoting science-based health decision-making, in this case, vaccine adherence.

Its role in the toolkit has been further articulated in Lines 673-678, as follows:

(3) increasing the use of brain-friendly (see previous section on Neuromarketing Science) vaccine promotion materials among priority populations. This specific role of Neuromarketing Science for content optimization is intended to arm communicators with confidence and feelings of trust crucial for making them effective communicators, delivering brain-friendly messaging to promote vaccine adherence.

This also has been done in Lines 696-701:

The third section, about Neuromarketing Science (NM), discusses how to apply brain-friendly messaging to develop vaccine education content. This section gives health communicators a basic understanding of Neuromarketing, the science of how the human brain processes information, along with specific practical tips for designing and delivering messaging that promotes a positive emotional response that facilitates beneficial cognitive processing of information.

Round 3

Reviewer 2 Report

Comments and Suggestions for Authors

The manuscript was improved as the text and results are more explicit. The rationale is adequate, and Table 2 was modified accordingly, as well as the text. I have no further observations